# Preparedness and Response to the COVID-19 Emergency: Experience from the Teaching Hospital of Pisa, Italy

**DOI:** 10.3390/ijerph17207376

**Published:** 2020-10-09

**Authors:** Angelo Baggiani, Silvia Briani, Grazia Luchini, Mauro Giraldi, Carlo Milli, Alfonso Cristaudo, Lucia Trillini, Lorenzo Rossi, Stefano Gaffi, Giovanni Ceccanti, Maria Carola Martino, Federica Marchetti, Marinella Pardi, Fabio Escati, Monica Scateni, Simona Frangioni, Antonella Ciucci, Guglielmo Arzilli, Daniele Sironi, Francesco Mariottini, Francesca Papini, Virginia Casigliani, Giuditta Scardina, Giacomo Visi, Costanza Bisordi, Tommaso Mariotti, Giulia Gemignani, Beatrice Casini, Andrea Porretta, Lara Tavoschi, Michele Totaro, Gaetano Pierpaolo Privitera

**Affiliations:** 1The Azienda Ospedaliero Universitaria Pisana, 56100 Pisa, Italy; silvia.briani@ao-pisa.toscana.it (S.B.); grazia.luchini@ao-pisa.toscana.it (G.L.); m.giraldi@ao-pisa.toscana.it (M.G.); c.milli@ao-pisa.toscana.it (C.M.); alfonso.cristaudo@unipi.it (A.C.); l.trillini@ao-pisa.toscana.it (L.T.); lo.rossi@ao-pisa.toscana.it (L.R.); s.gaffi@ao-pisa.toscana.it (S.G.); g.ceccanti@ao-pisa.toscana.it (G.C.); m.martino@ao-pisa.toscana.it (M.C.M.); f.marchetti@ao-pisa.toscana.it (F.M.); m.pardi@ao-pisa.toscana.it (M.P.); f.escati@ao-pisa.toscana.it (F.E.); m.scateni@ao-pisa.toscana.it (M.S.); s.frangioni@ao-pisa.toscana.it (S.F.); a.ciucci@ao-pisa.toscana.it (A.C.); g.gemignani@ao-pisa.toscana.it (G.G.); beatrice.casini@med.unipi.it (B.C.); andrea.porretta@unipi.it (A.P.); gaetano.privitera@unipi.it (G.P.P.); 2Department of Translational Research and New Technologies in Medicine and Surgery, University of Pisa, 56123 Pisa, Italy; g.arzilli3@studenti.unipi.it (G.A.); d.sironi@studenti.unipi.it (D.S.); francesco.mariottini@med.unipi.it (F.M.); f.papini1@studenti.unipi.it (F.P.); v.casigliani@studenti.unipi.it (V.C.); g.scardina@studenti.unipi.it (G.S.); g.visi1@studenti.unipi.it (G.V.); c.bisordi@studento.unipi.it (C.B.); 29397007@studenti.unipi.it (T.M.); Lara.tavoschi@unipi.it (L.T.); m.totaro2@studenti.unipi.it (M.T.)

**Keywords:** SARS-CoV-2, COVID-19 hospital, personal protective equipment, disinfection

## Abstract

In Italy, the coronavirus disease 2019 (COVID-19) emergency took hold in Lombardy and Veneto at the end of February 2020 and spread unevenly among the other regions in the following weeks. In Tuscany, the progressive increase of hospitalized COVID-19 patients required the set-up of a regional task force to prepare for and effectively respond to the emergency. In this case report, we aim to describe the key elements that have been identified and implemented in our center, a 1082-bed hospital located in the Pisa district, to rapidly respond to the COVID-19 outbreak in order to guarantee safety of patients and healthcare workers.

## 1. Introduction and Epidemiology

Italy has been heavily affected by coronavirus 2019 disease (COVID-19), along with other countries such as Spain [1]. During the pandemic, healthcare facilities had to quickly adapt to address the clinical complexity of COVID-19 cases, the high influx of cases in hospitals and the rise of transmission of the pathogen among hospitalized patients and Health Care Workers (HCW), modifying health service pathways greatly. In Italy, the first locally acquired COVID-19 cases were detected in the regions of Lombardy and Veneto on 21 February 2020 [2]. The country reached more than 100,000 cases and over 10,000 deaths by the end of March [3]. The severity of the outbreak convinced the Italian government to adopt unprecedented measures, with the imposition of a national lockdown on 9 March 2020 [4] and the closure of all activities not providing essential services on 22 March 2020 [5]. In Tuscany, an Italian region of 3.7 million inhabitants, the first confirmed case of COVID-19 was detected in Florence on 25 February 2020 [6]. Since then, the epidemic has grown rapidly, though at a lower rate than has been observed in the most affected Italian regions (i.e., Lombardy), and the epidemic has also grown unevenly among different provinces, with Massa and Lucca recording the highest impact. The number of cumulative cases increased significantly during March and April 2020, with an exponential growth at the start that later turned out to be better approximated by the Gombertz function, with the exponential growth slowing down and becoming linear until a plateau was reached. In Tuscany, the number of new daily cases was highest at the beginning of April (maximum n = 406 on 2 April) and then decreased gradually (<100 new cases on 21 April). The number of individuals simultaneously requiring hospital care reached its peak in the first week of April (n = 1149 on 3 April) and remained steady for a few weeks before slowly decreasing at the end of the month. During the plateau, individuals requiring intensive care represented almost 25% of inpatients (maximum hospitalized patients = 297 on 1 April). Since 24 April, 742 deaths have been reported in Tuscany among COVID-19 cases, with a case-fatality rate of 8.2%, lower than the national average of 13.4% [7]. Figure 1 reports the COVID-19 infection trend in Tuscany during the emergency.

On the same date, the north-west (NW) area was the most affected, followed by the central and the south-east (SE) areas. Additionally, hospitalization rates (HR) among cases and proportions of inpatients requiring intensive care varied greatly among different sub-regional areas, with the NW area presenting an HR of 10.2% (C: 15.4% and SE: 9.8%) and a proportion of patients admitted to intensive care units (ICUs) of 21.3% (C: 17.0% and SE: 16.4%).

## 2. Regional Management of Pandemic

Health services in Tuscany are organized by the regional department of health (RDH) in three sub-regional areas: central, with a population of 1,500,000 and 3000 hospital beds; NW, with a population of 1,200,000 and 3000 hospital beds; and SE, with a population of 800,000 and 1600 hospital beds. In each area there is one teaching hospital (TH), a number of district hospitals (DH) and smaller hospitals. On 4 March, the RDH established a regional task force (TF) with the aim of coordinating the preparedness plan of the regional health services and providing technical guidance [8]. The assessment of the demand for ICU beds in Tuscany was carried out by the Remote Center for Medical Relief Operations (RCMRO), founded in Pistoia (Tuscany, Italy). The RCMRO coordinated the availability and use of beds in hospitals outside of Tuscany.

THs were designated as the referral hospitals for patients affected by COVID-19 in their respective area (core hospital type A). DHs were divided into two groups: hospitals type B (secondary hospitals) and type C (primary hospitals). Type B hospitals provide care for time-dependent diseases like stroke, and a sufficient number of ICU beds are reserved for COVID-19 patients if needed. Type C hospitals, such as some public hospitals and all nursing homes, lack adequate ICU and are reserved for patients who tested negative for SARS-CoV-2. In anticipation of a high influx of ICU COVID-19 cases, planned activities were scaled down, resulting in an ICU occupation rate of 68% on March 19 [9]. In the NW area, more than half of ICUs (46 of 79) and ordinary beds (184 of 326) available for COVID-19 patients were housed in our tertiary hospital.

Other measures the territorial health care services took to respond to the additional needs included the creation of special units composed of a doctor and a nurse that provided dedicated home care to patients with suspected or confirmed COVID-19 infection; assessing the need of nasopharyngeal swab or hospitalization [10]; the set-up of health hotels for COVID-19 patients discharged from hospital but still positive; increasing the number of beds for intermediate care hospitals [11]; and reinforcing the epidemiological monitoring of nursing homes and long-term care facilities for elderly [12].

## 3. The North-West Area Setting and the Azienda Ospedaliero-Universitaria Pisana Renovation Plan

The NW area registered its first case on 27 February. A few days earlier, territorial healthcare services and the NW referral hospital, Azienda Ospedaliero-Universitaria Pisana (AOUP), began their reorganization. The AOUP is a highly specialized teaching, tertiary, 1082-bed hospital, and it is organized into two main facilities: Cisanello and Santa Chiara.

Shortly after the first Tuscany COVID-19 case, AOUP organized a multidisciplinary TF, composed of experts in infection prevention and control (IPC), occupational medicine and hospital management. Based on scientific data published since the start of the outbreak and previous experience from former outbreaks (SARS and MERS-CoV), the TF developed a technical procedure to face the pandemic, structured in five key domains: reorganization of hospital services, management of suspected or confirmed COVID-19 patients, management of corpses, guidelines for cleaning and disinfection, implementation of cleaning and disinfection procedures, and personal protective equipment.

### 3.1. Reorganization of Hospital Services

After drafting the procedure, the TF divided the Emergency Medicine Hospital into COVID-19 and not-COVID-19 areas, which were organized into accident and emergency (A&E), clinical wards, ICUs and operating rooms. Healthcare workers were assigned to one area or another with no interchangeability.

The A&E in COVID area was organized into four settings: pre-triage area, triage area, one negative pressure room and one room to perform the nasopharyngeal swab (swabbing room). A&E in not-COVID-19 area consisted of one triage area.

Clinical wards of COVID area were divided into infectious disease units and pulmonology units. From 19 March, several clinical wards and operating rooms, located in different hospital units, have been repurposed to realize 160 additional beds in COVID-19 clinical wards (COVID-19 area 1-2-3-4).

In ICUs of COVID-19 area, a total of 83 beds were divided into five ICUs (ICUs 1-5). Out of 83 beds, 53 (65%) were placed in a positive-pressure environment and may be considered as “conventional ICU beds”. The remaining 30 beds were installed in negative-pressure rooms. Negative-pressure isolation room system (−9Pa) and HEPA15 filters were installed to contain airborne microorganisms within the room. Negative pressure isolation rooms were used for COVID-19-positive patients with respiratory failure, usually treated with aerosol-generating C-PAP therapy (C-PAP beds) [13].

ICUs of not-COVID-19 areas were reduced to 38 beds. COVID-19 and not-COVID-19 areas included 3 and 23 operating rooms, respectively. Subdivision of COVID-19 and not-COVID-19 areas is shown in Figure 2, Figure 3 and Figure 4.

### 3.2. Management of Suspected or Confirmed COVID-19 Patients

In AOUP, a pre-triage tent was set up in front of the A&E entrance and a pre-triage nurse performed a symptoms-based screening of incoming patients, including those arriving by ambulance). Individuals with fever or respiratory symptoms, or those fulfilling any epidemiological criteria, were considered suspected cases [14]. Testing of suspected COVID cases was conducted by a dedicated triage nurse in one of the A&E COVID-19 area rooms; all other patients were directed to A&E not-COVID area triage. Dedicated HCWs went to the isolation room and made an evaluation of the clinical conditions. On the basis of anamnestic and clinical examination, the suspected case of COVID-19 could be excluded or confirmed. If confirmed, the patient underwent the nasopharyngeal swab for SARS-CoV-2. Patients who had emergency clinical signs and needed surgery or interventional procedures were subjected to swab in A&E or directly in the operating/interventional room. Since 9 March (the day national lockdown started), the procedure of admission changed and every patient who needed to be hospitalized was tested for COVID. If the patient’s clinical condition suggested the need for a radiological examination, a reserved path was defined.

Suspected COVID cases who needed an immediate CPAP cycle were treated in the A&E negative pressure box or were immediately hospitalized in a surgery room turned into a negative-pressure CPAP room. Until the swab result was available, the patient remained in a dedicated area. Upon arrival of the swab outcome, if the swab was positive, the patient was directed to the COVID area (ICU or medical ward). Otherwise, if the swab was negative, the patient was directed to the not-COVID area (ICU or medical ward).

Despite a negative swab, patients with strong clinical/radiological suspicion were directed to the COVID area (for these cases it was recommended to repeat the swab after 24 h). For COVID wards, a monitoring and management system called “Visual COVID” was established, allowing the real-time display of the beds available in areas of various care intensity.

### 3.3. Management of Corpses

A preparedness plan for the safe handling of dead bodies of suspected or confirmed COVID-19 cases at the site of death and during transport and storage in the hospital morgue was established in accordance with national institutional reports [15,16]. To secure safety for healthcare staff at the site of death, a face mask was placed over the mouth of the deceased in order to prevent the release of droplets. Staff responsible for transport of bodies from the site of death to the morgue had to wear appropriate PPE, and the corpses had to be wrapped in a sheet and a body bag (two if significant leakage of bodily fluids was present) in order to avoid contamination of environmental surfaces and minimize risk of transmission to staff related to direct contact with human remains.

Once arrived at the morgue, if the body had to be placed in the cold room, it was preferable to choose the lower cells, and it had to be written on the appropriate blackboard that the body was a COVID-19 dead body. Forensic examination of COVID corpses was discouraged if not strictly necessary. Before the closure of the coffin, one family member at a time was authorized to see the body from a distance of at least two meters.

### 3.4. Guidelines for Cleaning and Disinfection

Cleaning and disinfection procedures were drawn up in accordance with the available scientific indications issued by the main international health organizations [17,18,19]. Disinfection methods used for SARS-CoV-1 and MERS-CoV did not differ from those used routinely in the hospital [20] and were assumed to be reasonably valid also for SARS-CoV-2. As recommended by the Italian National Institute of Health, the following biocides were applied:70% ethyl alcohol;0.5% hydrogen peroxide;sodium hypochlorite (0.1–0.5% free chlorine);other disinfectants proved for virucidal activity according to European Standard (EN) 14,476 for medical devices [21].

### 3.5. Implementation of Cleaning and Disinfection Procedures

In AOUP, appropriately trained cleaning staff sanitized COVID-19 patients’ rooms four times per day. Moreover, after the discharge or transfer of a COVID-19 patient, the nursing coordinator initiated the cleaning service by filling a dedicated form. In addition, reusable medical devices were cleaned and properly sanitized. Laboratory equipment was sanitized in accordance with the manufacturer’s instructions or in accordance with the internal laboratory protocols. Disposable cloth with disinfect solution was used to reduce risk of aerosol spreading.

Floors and high touch surfaces (bed sides, bedside tables, tabletops, handles, push buttons and all bathroom surfaces) were cleaned with water and common detergents followed by disinfection with sodium hypochlorite (0.5% free chlorine). Sodium hypochlorite was used for biological liquid decontamination (15 min before removing and sanitizing it, using disposable wipes).

Following all activities involving aerosol production (e.g., bed preparation) and before proceeding with any other activities, rooms had to be adequately ventilated (at least 30 min in case of natural ventilation and if an adequate ventilation system was not available). In accordance with the above, used bed linen was placed in a hermetically sealed bag that was sent directly to an industrial washing chain (washing cycle of 60 °C for at least 30 min).

According to currently available data [22], wastes generated during patient care had to be managed according to current hospital protocols, without specific additional measures. For laboratory wastes, dedicated procedures have been applied.

Cleaning personnel wore appropriate personal protective equipment (PPE) described for service personnel (Table 1), including an additional pair of rubber gloves. All staff received appropriate training in dressing/undressing procedure.

### 3.6. Personal Protective Equipment (PPE)

In not-COVID-19 areas, patients and workers used surgical masks, gloves and gowns as PPE as described by WHO and National Institute of Health [23,24].

In COVID-19 areas, FFP2 or FFP3 masks, eye protection, a double pair of gloves and a second gown were recommended, mostly during the aerosol-generating procedures. All masks needed to be certified as described by British Standard (BS) EN 149:2001 standard [25].

All staff in COVID-19 and not-COVID-19 areas were trained on how to dress and undress. The dressing and undressing procedures for healthcare workers in COVID-19 areas followed the national guidelines [26].

## 4. Conclusions

This case report gives an overview of the strategies adopted in our teaching hospital to respond to the first phase of the emergency. The hospital response had to be integrated with a territorial response, where the creation of multidisciplinary teams composed of different professional figures secured the presence of medical assistance for suspected or confirmed COVID-19 patients outside the hospital.

In our opinion, among all possible measures that could be adopted by the hospital, the differentiation of COVID/not COVID pathways associated with the delay of all surgical interventions that were not urgent and the decision to limit hospital admission only to people needing immediate care were the most appropriate interventions to reduce transmission of the disease among other patients and healthcare staff.

As described elsewhere [27], when an outbreak of an emerging infectious disease occurs, it is important to use revised triage and hospital protocols to reorganize the provision of healthcare services. The experience of the AOUP can be useful for other similar tertiary hospitals facing this and future emergencies. Furthermore, the possibility of a second wave in our nation is not to be excluded, and many countries around the world are still struggling to control the emergency. 

## Figures and Tables

**Figure 1 ijerph-17-07376-f001:**
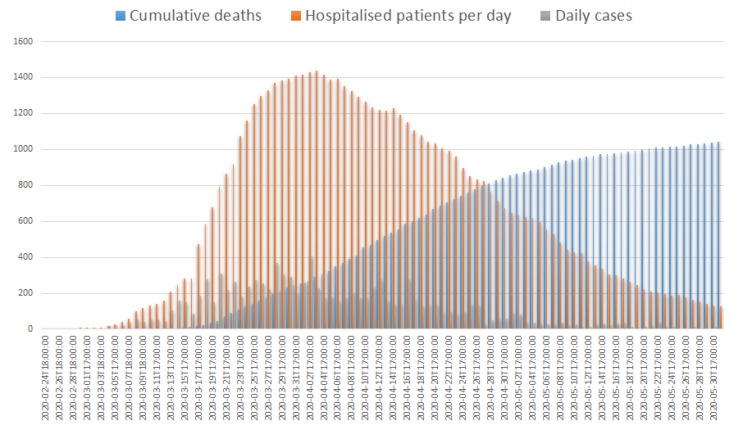
Trends of daily new cases, cumulative deaths and cumulative hospitalized patients per day, in Tuscany, from 24 February to 31 May.

**Figure 2 ijerph-17-07376-f002:**
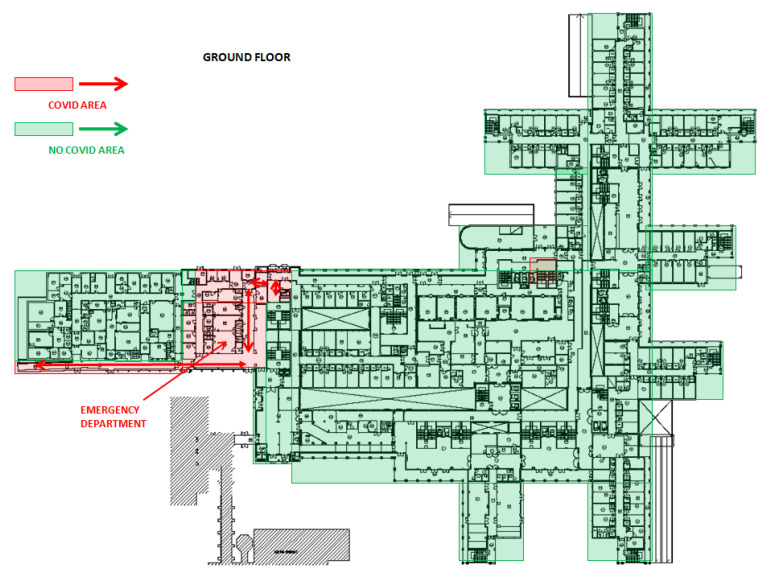
Planimetry of the ground floor of Cisanello hospital sector, where the COVID areas (in red) were set up.

**Figure 3 ijerph-17-07376-f003:**
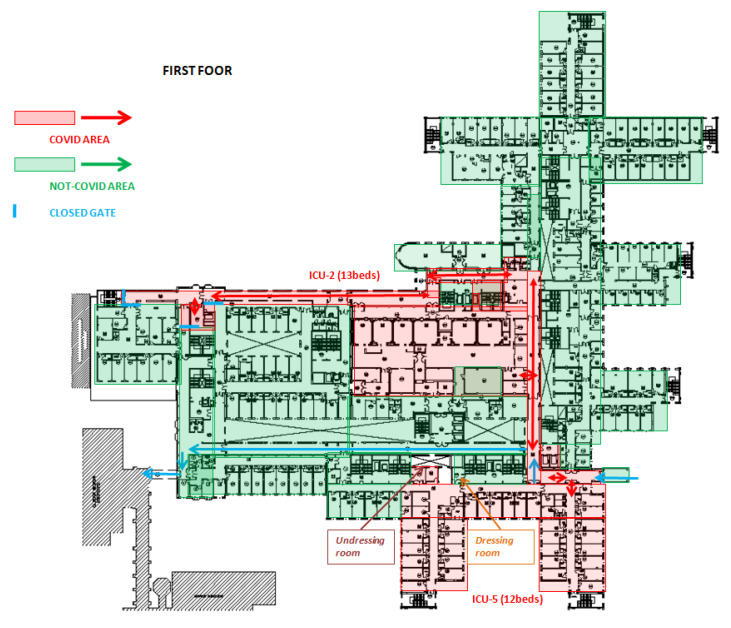
Planimetry of the first floor of Cisanello hospital sector, where the COVID areas (in red) were set up.

**Figure 4 ijerph-17-07376-f004:**
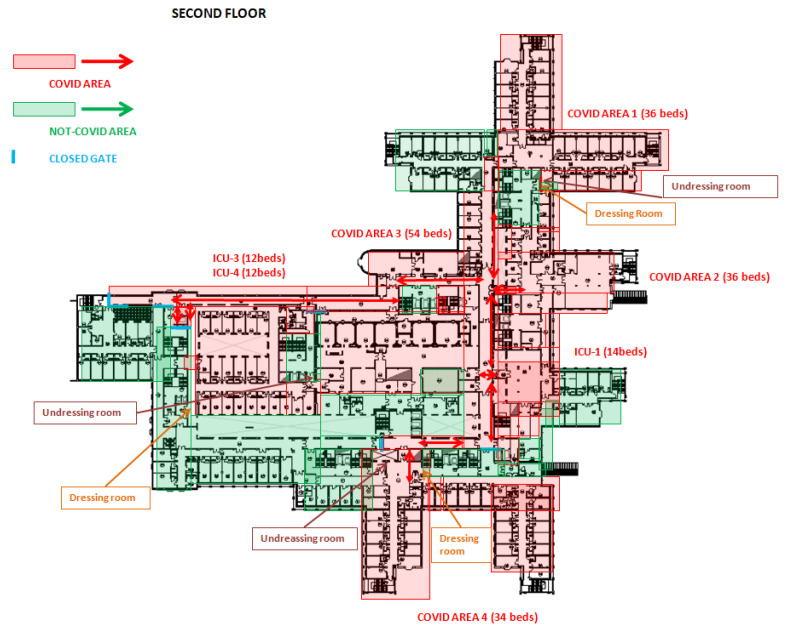
Planimetry of the second floor of Cisanello hospital sector, where the COVID areas (in red) were set up.

**Table 1 ijerph-17-07376-t001:** Rational use of personal protective equipment in Coronavirus infectious disease (COVID) and not-COVID areas.

**Setting 1—Emergency Department (Pre-Triage Area—Emergency Room—Diagnostic Radiology)** **All Patients Have Surgical Mask**							
**Location**	**Activity**	**Surgical Mask**	**FFP2/FFP3 Mask**	**Eye Protection**	**Gown**	**Gloves**	**Overshoes and Hair Cap**
Pre-Triage Area	Pre-Triage	X			X	1 pair	
Not-COVID Area	Triage, Healthcare, Cleaning, Reception	X			X	1 pair	
COVID Area	Swabbing, Healthcare, Cleaning		X	X	X	2 pair	X
**Setting 2—Medical Stays** **All Patients Have Surgical Mask**							
**Location**	**Activity**	**Surgical Mask**	**FFP2/FFP3 Mask**	**Eye Protection**	**Gown**	**Gloves**	**Overshoes and Hair Cap**
Not-COVID Area	Healthcare, Swabbing, Cleaning	X	X(swabbing)	X(swabbing)	X	1 pair	X
COVID Area	Healthcare, Swabbing, Cleaning		X	X	X	2 pair	X
**Setting 3—Operating Rooms** **All Patients Have Surgical Mask**							
**Location**	**Activity**	**Surgical Mask**	**FFP2/FFP3 Mask**	**Eye Protection**	**Gown**	**Gloves**	**Overshoes and Hair Cap**
Not-COVID Area	Patient Entrance, Intubation and Intervention, Cleaning	X			X	1 pair	X
COVID Area	Patient Entrance, Intubation and Intervention, Cleaning		X	X	X	2 pair	X
**Setting 4—Delivery Area** **All Patients Have Surgical Mask**							
**Location**	**Activity**	**Surgical Mask**	**FFP2/FFP3 Mask**	**Eye Protection**	**Gown**	**Gloves**	**Overshoes and Hair Cap**
Not-COVID Area	Patient Entrance, Delivery, Cleaning	X			X	1 pair	X
COVID Area	Patient Entrance, Delivery, Cleaning		X	X	X	2 pair	X
**Setting 5—Mortuary** **All Patients Have Surgical Mask**							
**Location**	**Activity**	**Surgical Mask**	**FFP2/FFP3 Mask**	**Eye Protection**	**Gown**	**Gloves**	**Overshoes and Hair Cap**
Death in Not-COVID Area	Assistance, Transport, Cleaning	X			X	1 pair	
Death in COVID Area	Assistance, Transport, Cleaning	X			X	2 pair	X
Necropsy Area for COVID Dead Body	Necropsy		X	X	x	2 pair	X

Filtering Face Piece 2, FFP2; Filtering Face Piece 3, FFP3.

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
