# Peer review of "Preparedness and Response to the COVID-19 Emergency: Experience from the Teaching Hospital of Pisa, Italy"

_ijerph, 2020, doi:10.3390/ijerph17207376_

Round 1

Reviewer 1 Report

I thank the authors for rigorously following my recommendations in the previous review round.

The manuscript is now clearer and easier to follow.

Some further comments:

As this manuscript is submitted as a case report, it should be explained in the title, abstract and main text why it is a case report.

I would suggest not using the term graphic for a figure in the figure heading.

The subheadings should be shortened and a mixture of bullet points and enumerations should be avoided.

Meaningful paragraphs are preferable to single phrases.

Please cite the manufacturers etc. of the ingredients used.

Some editing and correction of spelling and spacing mistakes are needed.

Reviewer 2 Report

I have read the new version of the manuscript as well as the responses of the authors to my comments and suggestions. My remarks have been addressed and the manuscript has been improved.

Two minor comments that could improve the clarity and quality of this case report:

  1. Please, revise the manuscript to create more paragraphs instead of different sentences. For instance, Section 3.3 could be one paragraph instead of four unconnected sentences. The same happens with Section 3.4 and Section 4.
  2. Regarding Section 3.4 and 3.5, about the cleaning and disinfection procedures, and the personal protective equipment, do you have some data of COVID-19 infection among healthcare workers between this period in your installations?

Author Response

This manuscript is a resubmission of an earlier submission. The following is a list of the peer review reports and author responses from that submission.

Round 1

Reviewer 1 Report

Comments to the Authors

The  Manuscript ID: ijerph-870411, Title: Drafting a technical procedure for the prevention of COVID-19 emergency in an Italian teaching hospital (Pisa, Tuscany): a renovation plan for SARS COV-2 risk management,reports on the development and implementation of a technical procedure for the rapid response to COVID-19 emergency in the teaching hospital in Pisa, Tuscany.

Although the overall topic is very timely and of great global relevance, there is very little useful information provided as a clear rationale and a detailed method are entirely missing. As it is presented, this manuscript resembles more a case report or a commentary than an original article. Due to the vast amount of typos and careless mistakes (already in the abstract!) including correct and consistent writing of dates and figures, I recommend that the manuscript is proofread by an English native speaker.

Some comments:

The title is too long. Also, I recommend using keywords that are not part of the title.

Please provide a state-of-the art abstract and avoid mentioning all these names of places.

Figure 1 is subdivided into a-c. However, an informative figure legend is not provided. Also, details of figure production it is not described in the methods section.

Spelling:

3.2. Taking Care COVID-19 Patients: „of” missing?

procedures for healthcare workers in COVID-19 areas is described: are!

What does this phrase mean? “Morge could host of deceaded COVID and not-COVID.” Did the authors mean “morgue”?

Full stop is missing in e.g. end of abstract or in “Forensic examination is discouraged if not strictly necessary”…!

A imitations section would be useful to summarize methodological etc. shortcomings of the applied research approach; practical implications of the study findings should be more concrete.

In conclusion, there are some major issues that limit readability and overall merits of the manuscript. I cannot recommend this article for publication in its current form although the topic is of great relevance to combat the outcome of the covid-19 pandemic.

Reviewer 2 Report

Dear authors,

Please find above attached my comments and suggestions about the manuscript.

Kind Regards
